# The impact of Mendelian sleep and circadian genetic variants in a population setting

Michael N. Weedon[1]*, Samuel E. Jones[1,2], Jacqueline M. Lane[3,4,5], Jiwon Lee[6], Hanna M. Ollila[2,7,8,9], Amy Dawes[1], Jess Tyrrell[1], Robin N. Beaumont[1], Timo Partonen[10], Ilona Merikanto[10,11], Stephen S. Rich[12], Jerome I. Rotter[13,14], Timothy M. Frayling[1], Martin K. Rutter[15,16], Susan Redline[6], Tamar Sofer[6], Richa Saxena[3,4,5], Andrew R. Wood[1]*

1 Genetics of Complex Traits, College of Medicine and Health, University of Exeter, Exeter, Devon, United Kingdom, 2 Institute for Molecular Medicine Finland (FIMM), HiLIFE, University of Helsinki, Helsinki, Finland, 3 Center for Genomic Medicine, Massachusetts General Hospital and Harvard Medical School, Boston, Massachusetts, United States of America, 4 Broad Institute, Cambridge, Massachusetts, United States of America, 5 Department of Anesthesia, Critical Care and Pain Medicine, Massachusetts General Hospital, Boston, Massachusetts, United States of America, 6 Division of Sleep and Circadian Disorders, Brigham and Women's Hospital, Harvard Medical School, Boston, Massachusetts, United States of America, 7 Center for Genomic Medicine, Massachusetts General Hospital, Boston, Massachusetts, United States of America, 8 Program in Medical and Population Genetics, Broad Institute, Cambridge, Massachusetts, United States of America, 9 Department of Psychiatry and Behavioral Sciences, School of Medicine, Stanford University, Stanford, California, United States of America, 10 Department of Public Health Solutions, Finnish Institute for Health and Welfare, Helsinki, Finland, 11 SleepWell Research Program Unit, Faculty of Medicine, University of Helsinki, Helsinki, Finland, 12 Center for Public Health Genomics, Department of Public Health Sciences, University of Virginia, Charlottesville, Virginia, United States of America, 13 Institute for Genomics and Population Sciences, The Lundquist Institute for Biomedical Innovation, Torrance, California, United States of America, 14 Department of Pediatrics, Harbor-UCLA Medical Center, Torrance, California, United States of America, 15 Division of Diabetes, Endocrinology and Gastroenterology, Faculty of Medicine, Biology and Health, University of Manchester, Manchester, United Kingdom, 16 Diabetes, Endocrinology and Metabolism Centre, Manchester University NHS Foundation Trust, Manchester Academic Health Science Centre, Manchester, United Kingdom

* M.N.Weedon@exeter.ac.uk (MNW); A.R.Wood@exeter.ac.uk (ARW)

## Abstract

Rare variants in ten genes have been reported to cause Mendelian sleep conditions characterised by extreme sleep duration or timing. These include familial natural short sleep (*ADRB1*, *DEC2/BHLHE41*, *GRM1 and NPSR1*), advanced sleep phase (*PER2*, *PER3*, *CRY2*, *CSNK1D* and *TIMELESS*) and delayed sleep phase (*CRY1*). The association of variants in these genes with extreme sleep conditions were usually based on clinically ascertained families, and their effects when identified in the population are unknown. We aimed to determine the effects of these variants on sleep traits in large population-based cohorts. We performed genetic association analysis of variants previously reported to be causal for Mendelian sleep and circadian conditions. Analyses were performed using 191,929 individuals with data on sleep and whole-exome or genome-sequence data from 4 population-based studies: UK Biobank, FINRISK, Health-2000-2001, and the Multi-Ethnic Study of Atherosclerosis (MESA). We identified sleep disorders from self-report, hospital and primary care data. We estimated sleep duration and timing measures from self-report and accelerometery data. We identified carriers for 10 out of 12 previously reported pathogenic variants for 8 of the 10 genes. They ranged in frequency from 1 individual with the variant in *CSNK1D* to

meet the criteria for access to datasets to UK Biobank (http://www.ukbiobank.ac.uk).

**Funding:** M.N.W. was supported by grant MR/M005070/1 from Medical Research Council. A.R.W is supported by the Academy of Medical Sciences / the Wellcome Trust / the Government Department of Business, Energy and Industrial Strategy / the British Heart Foundation / Diabetes UK Springboard Award [SBF006\1134]. R.N.B is supported by grant MR/T00200X/1 from Medical Research Council. Whole-genome sequencing for "NHLBI TOPMed: Multi-Ethnic Study of Atherosclerosis (MESA)" was supported by grant phs001416.v1.p1 from the National Heart, Lung, and Blood Institute. The MESA study was supported in part by the National Center for Advancing Translational Sciences, CTSI grant UL1TR001881, and the National Institute of Diabetes and Digestive and Kidney Disease Diabetes Research Center (DRC) grant DK063491 to the Southern California Diabetes Endocrinology Research Center. The funders had no role in study design, data collection and analysis, decision to publish, or preparation of the manuscript.

**Competing interests:** The authors have declared that no competing interests exist.

1,574 individuals with a reported variant in the *PER3* gene in the UK Biobank. No carriers for variants reported in *NPSR1* or *PER2* were identified. We found no association between variants analyzed and extreme sleep or circadian phenotypes. Using sleep timing as a proxy measure for sleep phase, only *PER3* and *CRY1* variants demonstrated association with earlier and later sleep timing, respectively; however, the magnitude of effect was smaller than previously reported (sleep midpoint ~7 mins earlier and ~5 mins later, respectively). We also performed burden tests of protein truncating (PTVs) or rare missense variants for the 10 genes. Only PTVs in *PER2* and *PER3* were associated with a relevant trait (for example, 64 individuals with a PTV in *PER2* had an odds ratio of 4.4 for being "definitely a morning person", $P = 4\times10^{-8}$; and had a 57-minute earlier midpoint sleep, $P = 5\times10^{-7}$). Our results indicate that previously reported variants for Mendelian sleep and circadian conditions are often not highly penetrant when ascertained incidentally from the general population.

## Author summary

Clinically ascertained family-based studies have previously identified rare genetic variation associated with causing life-long sleep conditions, specifically shorter sleep, and earlier or later sleep timing. However, the effects of previously reported genetic variants on sleep duration and timing when identified incidentally through population-based studies are not known. Here, we take advantage of up to 191,929 individuals from four population-based studies, including the UK Biobank, to estimate the effects of these variants on sleep duration and timing using self-reported and accelerometer-based sleep estimates coupled with sequencing data. Our analysis revealed no association between variants previously reported and extreme sleep conditions. Two variants located in two genes (*PER3* and *CRY1*) showed evidence of association with sleep timing, but their estimated effects (~5 to 7 minutes) on sleep timing are much smaller relative to those previously reported. Our results indicate that previously reported variants are not causal for extreme sleep conditions in the general population. Finally, although we were unable to analyse a previously reported variant in the *PER2* gene associated with sleep timing, additional analysis in the UK Biobank revealed carries of protein-truncating variants in this gene have an approximately 1-hour earlier sleep midpoint compared to non-carriers. These population-based estimates are important because of the recent dramatic increase in direct-to-consumer and health service genome-wide genetic testing.

## Introduction

Rare variants in ten genes have been reported to cause Mendelian sleep conditions that are characterised by extreme sleep duration or timing. For example, variants in the *ADRB1*, *NPSR1* and *GRM1* genes have been recently reported to cause familial natural short sleep among carriers, defined as 4 to 6 hours sleep with no adverse effects on mental health or well-being[1–3]. Short sleep duration has also been reported to be caused by variants in the *DEC2/BHLHE41* gene, a well-known Mendelian sleep gene[4]. Familial advanced sleep phase (FASP) where sleep timing is shifted 3 or 4 hours earlier has been reported for variants in *PER2*[5], *PER3*[6], *CRY2*[7], *CSNK1D*[8] and *TIMELESS*[9]. The opposite condition, familial delayed sleep phase disorder has been reported to be caused by a gain-of-function *CRY1* variant,

c.1657+3A>C, with affected individuals sleeping approximately 1 hour later than unaffected individuals[10].

The effect of variants in these genes on sleep duration and sleep timing when identified incidentally in the general population is unknown. Discovery efforts for these variants generally used either a single pedigree or a small number of families selected on a specific clinical phenotype. For example, for *ADRB1*, six individuals in a single family were affected with familial natural short sleep. This "phenotype first" method of discovery means we do not know the effect of these variants when identified in an individual from the general population (i.e. from a "genotype first" approach). It is important to re-evaluate effect estimates to understand the underlying biology which may inform clinical risk stratification, and because of the recent dramatic increase in direct-to-consumer (DTC) and health service genome-wide genetic testing. To assess the effect of these variants when identified incidentally, large, unselected population cohorts are needed.

Estimating the effects of these variants in the general population has not previously been possible due to limitations in the availability of genetic data coupled with sleep parameters. The UK Biobank, a population-based study of 500,000 individuals from the UK, provides an opportunity to address questions of pathogenicity and penetrance of rare genetic conditions [11]. We have previously shown, for a range of traits and diseases, that disease penetrance is generally lower in UK Biobank compared to that reported from clinical cohorts[12,13]. For example, using activity monitor derived and self-report estimates of sleep timing from the UK Biobank, we have demonstrated that the effect of the *PER3* P415A/H417R familial advanced sleep phase variant on sleep timing is substantially lower than the published estimate (0.13hrs vs. 4.2hrs)[14]. However, our previous studies were based on genotyping array data of relatively common single nucleotide polymorphisms (SNPs). Genotyping arrays are known to capture a relatively small number of rare coding pathogenic disease variants, typically with poor accuracy for genotyping and imputation [13,15,16].

In this study, we use exome sequencing data in up to 184,065 individuals of European ancestry from the UK Biobank (October 2020 release) with sleep data, with additional data from up to 2,015 individuals from the Multi-Ethnic Study of Atherosclerosis (MESA), and 5,929 individuals from the FINRISK and Health 2000-2011 studies to comprehensively assess the penetrance of Mendelian sleep and circadian genes in a population-based setting. We show that previously reported variants for Mendelian sleep and circadian conditions are often not highly penetrant when identified incidentally from the general population.

## Methods

### Ethics statement

The UK Biobank was granted ethical approval by the North West Multi-centre Research Ethics Committee (MREC) to collect and distribute data and samples from the participants (http://www.ukbiobank.ac.uk/ethics/) and covers the work in this study, which was performed under UK Biobank application numbers 9072 and 16434. All participants included in these analyses gave written consent to participate.

### UK Biobank study participants

The primary study population was drawn from the UK Biobank study–a longitudinal population-based study of individuals residing in the UK. We restricted our analysis to a subset of 184,532 Europeans with whole-exome sequence data, including 170,518 unrelated Europeans (<3rd degree) defined through kinship coefficients made available from the UK Biobank. Details on derivation of genetic ancestry has previously been reported in Jones *et al.*[14].

## Exome sequence data in UK Biobank

We used the second release of exome-sequence data from the UK Biobank (October 2020). Specifically, we used genotypes called and provided in binary PLINK format (data field: 23155). Genotypes for previously reported Mendelian causes of sleep and circadian conditions were extracted for subsequent data analysis. Details of central processing of whole-exome data on 200K UK Biobank participants can be found online as part of UK Biobank's data showcase: https://biobank.ctsu.ox.ac.uk/crystal/label.cgi?id=170. All variants passed central quality control[17]. Sequence data for variants analysed in this paper were manually inspected through IGV[18] plots.

## Sleep phenotypes and variant selection

We focussed our analysis on phenotypes previously reported to have Mendelian causes, specifically familial natural short sleep (≤6 hours)[1,3,4], familial advanced sleep phase characterised by earlier sleep onset and earlier waking[5–9], and delayed sleep phase disorder associated with later sleep onset and later waking[10].

## Sleep disorders and medication

We used data from the UK Biobank variable 131061 (https://biobank.ndph.ox.ac.uk/showcase/field.cgi?id=131061) which classifies individuals as having a sleep disorder based on self-report, hospital and primary care data. As this variable does not separate out sub classifications of sleep disorders we used ICD-10 codes G472 and F512 from in-patient data to specifically assess disorders of the sleep wake cycle. Medication use at baseline in the UK Biobank was identified from variable 20003 (https://biobank.ctsu.ox.ac.uk/crystal/field.cgi?id=20003). Classification of sleep medication used in this paper has previously been described[19]. Since 'sleep' is a behavior, phenotyping is complicated. There are extreme sleep patterns like ASP and NSS which are often not considered 'disorders' by individuals if the trait does not interfere with an individual's work and social demands[20].

**Sleep duration.** We used self-reported sleep duration from UK Biobank questionnaire data (data field: 1160). We excluded individuals who reported >12 hours sleep duration, did not know or preferred not to answer. This data was also dichotomized for multiple analyses to define "short sleepers" as individuals self-reporting sleep duration ≤6 hours, ≤5 hours, ≤4 hours, and ≥4 hours ≤6. A maximum of 166,360 individuals had genotype and self-report sleep duration available across the variants previously reported to be Mendelian causes of familial natural short sleep and prioritised for analysis. In addition, we used accelerometer estimates of nocturnal sleep derived in a previous study[21]. A maximum subset of 34,241 individuals remained for analysis of accelerometer-based sleep duration estimates after removing individuals (n = 4,323) with problematic accelerometer data processing or who were outliers for the number of nocturnal sleep episodes used to derive nocturnal sleep estimates[21]. There was no association for individual variants or overall with the individuals removed from the accelerometer analyses. We also applied additional quality control among non-carriers of previously reported variants described in this study by removing individuals with >12 hours estimated sleep duration.

**Sleep timing.** We used self-reported chronotype (diurnal preference) available in the UK Biobank (data field: 1180) as a proxy for sleep timing, whereby we assumed morning people to sleep earlier and evening people to sleep later. We created four binary variables to represent chronotype where individuals were coded '1' based on being: 1) definitely a morning person; 2) more or definitely a morning person; 3) definitely an evening person, and 4) more or definitely an evening person. For each variable, individuals who did not report having the

respective circadian preference(s) (including "Do not know" but excluding "Prefer not to answer") were coded '0'. A maximum of 168,409 individuals had genotype data and self-report chronotype data across the variants associated with advanced and delayed sleep phase. In addition, we used accelerometer estimates from up to 7 nights of the least-active 5 hours (L5) over a 24-hour period, with values representing hours from the previous midnight (e.g. 7 p.m. = 19 and 2 a.m. = 26)[21]. Sleep midpoint was estimated as the mid-point of the sleep period time window used to define sleep duration[21]. In total, a maximum subset of 34,650 individuals remained for statistical analysis of accelerometer-based sleep timing estimates. We applied additional quality control among non-carriers of previously reported variants described by removing individuals outside 4 standard deviations of the respective trait (midpoint sleep or L5 timing) analysed.

## Replication of findings

To replicate our UK Biobank observations of previously reported variant-phenotype associations, we used self-reported measures of sleep duration and circadian preference (chronotype) in up to 5,929 individuals from two population-based studies from Finland: FINRISK[22] and Health-2000-2011 (https://www.julkari.fi/handle/10024/130780) and publicly available summary statistics from release 6 of FinnGen (https://r6.finngen.fi/). In addition, we used accelerometer-based estimates of sleep timing (sleep midpoint) in up to 1,935 individuals from the Multi-Ethnic Study of Atherosclerosis (MESA) Sleep Study[23] where sequence-based genotypes of previously reported variants associated with sleep timing from its Exam 5 were available (see S1 Methods for study descriptors).

## Statistical Analysis

**Defining genotype groups for comparison of sleep parameter estimates.** For each variant, summaries of sleep parameter estimates were analysed by genotype. The set of individuals classified as homozygous reference for all variants was the same after removing individuals from this genotype group who were carriers for any previously reported variant.

**Analysis of self-report phenotypes.** t-tests were performed to compare means and standard deviations of the continuous self-report sleep duration variables across genotype groups and carrier status, respectively. For dichotomized variables, Fisher's exact tests were performed to compare proportions of individuals labelled as short sleepers or had a defined circadian preference (above) across genotype groups. Alternate homozygous counts were combined with heterozygous counts when performing Fisher's exact test where applicable. In addition, logistic regression was performed using UK Biobank data to obtain odds ratios, adjusting for age at baseline (field 21003), sex (field 31), assessment centre (field 54), month when attending the assessment centre (field 55), and 40 genetic principal components (field 22009) available from the UK Biobank.

**Accelerometer-derived phenotypes.** t-tests were performed to compare means and standard deviations of the accelerometer-based estimates of sleep duration, sleep-midpoint and L5 timing.

**Burden testing of rare loss-of-function and missense variants in UK Biobank.** Genetic variants identified in the ten previously reported genes were annotated using the Ensembl Variant Effect Predictor (VEP)[24] and LOFTEE[25]. Variants with MAF<0.0001 and annotated as missense or loss-of-function with high confidence were analysed through burden testing as implemented in REGENIE[26] that accounts for relatedness among individuals analysed. We analysed up to 184,065 individuals of inferred European genetic ancestry with available sleep and covariate data.

## Results

### Most reported Mendelian sleep variants are present in UK Biobank and are not associated with sleep disorders

We assessed the frequencies of 12 variants in 10 genes that have been reported to cause familial natural short sleep, familial advanced sleep phase, or delayed sleep phase conditions (S1 Table). All were present in the UK Biobank in unrelated individuals of European ancestry except for the S662G variant in *PER2* (not present in gnomAD (version 2.1)) and the Y206H variant in *NPSR1*. In addition, we identified carriers for previously reported variants in *GRM1*, *BHLHE41/DEC2*, *CRY1*, and *PER3* in the MESA and Finnish studies (S2 Table). None of the variants were associated with any self-report or clinically diagnosed sleep disorder in UK Biobank, including the G472 or F512 ICD-10 code for disorders of the sleep wake cycle or with sleep medication use (Table 1). We noted that in release 6 of the FinnGen study there was a nominal association with ICD10 code G472 (circadian rhythm sleep disorders) for the *CRY1* c.1657+3A>C variant (0.05% in controls vs. 0.29% in cases, *P* = 0.026), but this variant is 10-fold rarer in the Finnish population than in non-Finnish Europeans and is imputed with a quality score of only 0.76.

**Table 1. Sleep disorder and medication status by carrier status of variants previously reported to affect sleep duration or timing.** Sleep disorders in the UK Biobank coded as G47 from self-report, ICD-10 codes or primary care data. Also presented are results for disorders of the sleep wake cycle (ICD10: G472, F512). Sleep medication codes reported at UK Biobank baseline at reported in Lane *et al*. Nature Genetics, 2019.

| Gene | Variant | Genotype | Any self-report, primary care or ICD-10 code sleep disorder record (G47) | | | | Any sleep medication | | | | Any ICD-10 code for sleep-wake disorder (G472 or F512) | | | |
|---|---|---|---|---|---|---|---|---|---|---|---|---|---|---|
| | | | Controls N | Cases N | Cases% | Pª | No sleep Meds N | Sleep Meds N | Sleep Meds % | Pª | Controls N | Cases N | Cases % | Pª |
| *ADRB1* | A187V | C/C | 164,526 | 5,921 | 3.47 | 0.310 | 155,977 | 14,467 | 8.49 | 0.386 | 170,420 | 27 | 0.02 | 1.000 |
| | | C/T | 66 | 4 | 5.71 | | 62 | 8 | 11.43 | | 70 | 0 | 0.00 | |
| *DEC2/BHLHE41* | P384R | G/G | 164,577 | 5,921 | 3.47 | 0.298 | 156,023 | 14,472 | 8.49 | 0.588 | 170,471 | 27 | 0.02 | 1.000 |
| | | G/C | 9 | 1 | 10.00 | | 9 | 1 | 10.00 | | 10 | 0 | 0.00 | |
| *GRM1* | S458A | T/T | 164,526 | 5,923 | 3.47 | 1.000 | 155,976 | 14,470 | 8.49 | 1.000 | 170,422 | 27 | 0.02 | 1.000 |
| | | T/G | 65 | 2 | 2.99 | | 62 | 5 | 7.46 | | 67 | 0 | 0.00 | |
| | A889T | A/A | 164,586 | 5,925 | 3.47 | 1.000 | 156,033 | 14,475 | 8.49 | 1.000 | 170,484 | 27 | 0.02 | 1.000 |
| | | A/T | 3 | 0 | 0.00 | | 3 | 0 | 0.00 | | 3 | 0 | 0.00 | |
| *PER3* | P415A | C/C | 163,079 | 5,862 | 3.47 | 0.240 | 154,589 | 14,348 | 8.49 | 0.413 | 168,914 | 27 | 0.02 | 1.000 |
| | | C/G | 1,505 | 62 | 3.96 | | 1,442 | 123 | 7.86 | | 1,567 | 0 | 0.00 | |
| | | G/G | 6 | 1 | 14.29 | | 6 | 1 | 14.29 | | 7 | 0 | 0.00 | |
| | H417R | A/A | 163,078 | 5,862 | 3.47 | 0.268 | 154,589 | 14,348 | 8.49 | 0.525 | 168,913 | 27 | 0.02 | 1.000 |
| | | A/G | 1,507 | 62 | 3.95 | | 1,444 | 125 | 7.97 | | 1,569 | 0 | 0.00 | |
| | | G/G | 6 | 1 | 14.29 | | 6 | 1 | 14.29 | | 7 | 0 | 0.00 | |
| *CRY2* | A260T | G/G | 164,555 | 5,924 | 3.47 | 1.000 | 156,008 | 14,468 | 8.49 | 0.044 | 170,452 | 27 | 0.02 | 1.000 |
| | | G/A | 38 | 1 | 5.71 | | 32 | 7 | 17.95 | | 39 | 0 | 0.00 | |
| *TIMELESS* | R1081X | G/G | 164,580 | 5,925 | 3.47 | 1.000 | 156,027 | 14,475 | 8.49 | 1.000 | 170,478 | 27 | 0.02 | 1.000 |
| | | G/A | 6 | 0 | 5.71 | | 6 | 0 | 0.00 | | 6 | 0 | 0.00 | |
| *CRY1* | c.1657+3A>C | T/T | 163,148 | 5,879 | 3.47 | 0.477 | 154,669 | 14,355 | 8.49 | 0.575 | 169,000 | 27 | 0.02 | 1.000 |
| | | T/G | 1,435 | 46 | 3.47 | | 1,362 | 119 | 8.04 | | 1,481 | 0 | 0.00 | |
| | | G/G | 9 | 0 | 5.71 | | 8 | 1 | 11.11 | | 9 | 0 | 0.00 | |
| *CSNK1D* | T44A | T/T | 164,592 | 5,925 | 3.47 | 1.000 | 156,039 | 14,475 | 8.49 | 1.000 | 170,490 | 27 | 0.02 | 1.000 |
| | | T/C | 1 | 0 | 5.71 | | 1 | 0 | 0.00 | | 1 | 0 | 0.00 | |

[a]P-value derived from 2-sided Fisher's exact test. Homozygous carriers for minor alleles were combined with heterozygous carriers prior to performing Fisher's exact test.

### *ADRB1*, *GRM1*, or *DEC2/BHLHE41* pathogenic variants are not associated with self-reported short sleep duration in population-based cohorts

We identified 149 unrelated individuals from the UK Biobank with self-reported measures of sleep duration and carrying a previously reported pathogenic variant for natural short sleep in *ADRB1 (*A817V, *n* = 69*)*, *DEC2/BHLHE41 (*P384R, *n* = 10*)*, or *GRM1 (*S458A, *n* = 67; A889T, *n* = 3)*. We found no evidence that these individuals have short sleep durations in the UK Biobank (**Tables 2**, **S3** and **S4**). For example, the 69 carriers of the *ADRB1* variant had a self-reported average sleep duration of 7.1 hours (95% CI: 6.9, 7.3) compared to 7.2 hours (95% CI: 7.19, 7.21) among non-variant carriers (t-test *P* = 0.62). The proportion of *ADRB1* variant carriers self-reporting ≤6 hours of sleep, 23.2%, was no different to the proportion of people not carrying the variant, 23.7% (Fisher's exact *P* = 1.00). Using a fully adjusted logistic regression model gave similar results with an odds ratio of 0.98 (95% CI: 0.56–1.72; *P* = 0.96) for sleeping ≤ 6 hours. We observed zero carriers of the *ADRB1* variant self-reporting a more extreme phenotype of ≤4 hours of sleep. Similar observations were made for previously reported monogenic sleep disruption variants in *DEC2/BHLHE41* and *GRM1*. The 10 carriers of the P384R variant in the *DEC2/BHLHE41* gene had an average self-reported sleep duration of 7.3 hours compared to 7.2 hours among non-carriers (*P* = 0.69), with 10% of carriers reporting ≤6 hours sleep duration compared to 23.7% among non-carriers of reported variants (P = 0.47). Carriers for variants in the *GRM1* gene did not significantly differ in sleep duration from non-carriers (S458A: 7.1 hours (carriers) vs 7.2 hours (non-carriers), P = 0.81; A889T: 8.0 hours (carriers) vs 7.2 hours (non-carriers), P = 0.18), or the number of individuals reporting sleep duration of ≤6 hours (S458A: 28.4% (carriers) vs 23.7% (non-carriers), P = 0.39; A889T: 0% (carriers) vs 23.8% (non-carriers), P = 1.00). The null effect of the S458A variant in *GRM1* was also observed in FINRISK/Health 2000-2011 where <5 S458A carriers were

**Table 2. Summary of self-reported sleep duration in UK Biobank (data field 1160) between carriers of variants previously reported to be causal for familial natural short sleep and non-carriers (homozygous reference called by UK Biobank exome-sequencing) who are also non-carriers for any of the other 12 variants described in this article.**

| Gene | Variant | REF/ALT[a] | Study | Genotype | N[b] | Min[c] | Max[d] | Self-report sleep duration (hrs) | | Reporting ≤6 hours sleep | | |
| | | | | | | | | Mean (SD[e]) | P[f] | %[g] | Cases / Controls | P[h] |
|---|---|---|---|---|---|---|---|---|---|---|---|---|
| *ADRB1* | A187V | C/T | UKB | C/C | 166,291 | 1 | 12 | 7.17 (1.07) | 0.62 | 23.74 | 39,484 / 126,807 | 1.00 |
| | | | | C/T | 69 | 5 | 12 | 7.10 (1.03) | | 23.19 | 16 / 53 | |
| *DEC2/BHLHE41* | P384R | G/C | UKB | G/G | 166,283 | 1 | 12 | 7.17 (1.07) | 0.69 | 23.74 | 39,483 / 126,800 | 0.47 |
| | | | | G/C | 10 | 5 | 9 | 7.30 (1.06) | | 10.00 | 1 / 9 | |
| *GRM1* | S458A | T/G | UKB | T/T | 166,290 | 1 | 12 | 7.17 (1.07) | 0.81 | 23.74 | 39,484 / 126,806 | 0.39 |
| | | | | T/G | 67 | 4 | 10 | 7.13 (1.15) | | 28.36 | 19 / 48 | |
| | | | FINRISK/Health 2000-2011 | T/T | 5,927 | 3 | 15 | 7.59 (1.20) | 0.92 | 14.10 | 837 / 5,090 | 1.00 |
| | | | | T/G | <5 | 7 | 8 | 7.50 (0.71) | | 0.00 | 0 / <5 | |
| | A889T | A/T | UKB | A/A | 166,288 | 1 | 12 | 7.17 (1.07) | 0.18 | 23.75 | 39,485 / 126,803 | 1.00 |
| | | | | A/T | 3 | 8 | 8 | 8.00 (0.00) | | 0.00 | 0 / 3 | |

[a]reference allele / alternate allele

[b]number of individuals in genotype group

[c]minimum sleep duration self-reported in genotype group

[d]maximum sleep duration self-reported in genotype group

[e]standard deviation

[f]t-test *P*-value (two-sided)

[g]% = percentage of individuals in genotype group self-reporting ≤6 hours sleep duration (cases)

[h]Fisher's exact-test *P*-value (two-sided).

identified and had no statistically significant difference compared to non-carriers in average sleep duration (t-test $P = 0.92$) or proportion self-reporting $\leq 6$ hours sleep (Fisher's exact $P = 1.00$) (**Table 2**).

### *ADBR1*, *GRM1*, or *DEC2/BHLHE41* pathogenic variants are not associated with accelerometer derived measures of sleep in population-based cohorts

We confirmed the lack of association between previously identified pathogenic variants and sleep duration using accelerometer estimates of sleep in a subset of 34,226 individuals from the UK Biobank. Fifteen *ADRB1* variant carriers with accelerometer-derived sleep estimates had an average sleep duration of 7.6 hours (95% CI: 7.4, 7.8) compared to 7.3 hours (95% CI: 7.29, 7.31) among non-variant carriers (t-test P = 0.20) (**Table 3**). All 15 *ADRB1* variant carriers had an accelerometer-based sleep duration average of more than 6 hours (min = 6 hours, 53 minutes). Similar observations were made when stratifying accelerometer data analyses to weekend nights and weekday nights (**S5 Table**).

### *PER3*, but not *CRY2* or *TIMELESS*, variants are associated with advanced sleep phase in the population-based cohorts, but with reduced effect size

Variants in five genes have previously been associated with familial advanced sleep phase syndrome–characterised by approximately $\geq 3$ hour shifts towards earlier sleep and wake times. We previously tested the *PER3* P415A/H417R variant in the UK Biobank and found it was associated with chronotype and activity monitor derived sleep timing, although the size of the effect on sleep timing (L5 time) was smaller than the initially published estimate of 4.2 hours (7.8 minutes, 95% CI: 4.2–13.2 minutes, $P = 4.3 \times 10^{-4}$)[14]. We confirmed this association using exome sequence data and accelerometer data. The difference on average sleep-midpoint timing between carriers and non-carriers based on exome-sequence data was 6.8 minutes (95% CI: 1.4–12.3 minutes, $P = 0.01$) with a similar effect size for L5 timing (**Table 4**). Variant carriers had an odds ratio of 1.36 (95% CI: 1.22–1.52, P = $2 \times 10^{-8}$) for "definitely" being a morning person. We found no evidence that carriers of previously reported pathogenic variants in the other two genes, *CRY2* and *TIMELESS*, had altered chronotype, L5 timing or sleep-midpoint indicative of earlier sleep timing (**S6–S8 Tables**).

**Table 3. Summary of average accelerometer derived sleep duration (hours) (all nights) in the subset of exome-sequenced unrelated Europeans in UK Biobank, split by carriers and non-carriers for variants previously reported to be causal for familial natural short sleep.**

| Gene | Variant | Genotype | N[a] | Min[b] | Max[c] | Mean | SD[d] | P[e] |
|------|---------|----------|------|--------|--------|------|-------|------|
| *ADRB1* | A187V | C/C | 34,168 | 1.63 | 11.87 | 7.30 | 0.86 | 0.20 |
| | | C/T | 15 | 6.89 | 8.83 | 7.59 | 0.49 | |
| *DEC2/BHLHE4* | P384R | G/G | 34,167 | 1.63 | 11.87 | 7.30 | 0.86 | 0.62 |
| | | G/C | 4 | 6.26 | 8.25 | 7.51 | 0.92 | |
| *GRM1* | S458A | T/T | 34,166 | 1.63 | 11.87 | 7.30 | 0.86 | 0.92 |
| | | T/G | 10 | 6.48 | 8.30 | 7.33 | 0.76 | |
| | A889T | A/A | 34,167 | 1.63 | 11.87 | 7.30 | 0.86 | - |
| | | A/T | 0 | - | - | - | - | |

[a]number of individuals in genotype group

[b]minimum average sleep duration in genotype group

[c]maximum average sleep duration in genotype group

[d]standard deviation

[e]t-test *P*-value (two-sided).

**Table 4. Proportion of individuals self-reporting as being "definitely a morning person", average L5 timing, and average of sleep-midpoint across all nights for each genotype group of variants previously reported to be causal for familial advanced sleep phase in the UK Biobank (UKB), Finnish and MESA studies.** Accelerometer-based estimates of sleep timing unavailable in the Finnish studies. Self-reported "morningness" and accelerometer estimates of L5-timing unavailable in MESA.

| Gene | Variant | Study | Genotype | Definitely a "Morning" Person | | | Accelerometer L5 Timing | | | | | | Accelerometer Sleep Midpoint Timing | | | | | |
|---|---|---|---|---|---|---|---|---|---|---|---|---|---|---|---|---|---|---|
| | | | | %[a] | Cases / Controls | P[b] | N[c] | Min[d] | Max[e] | Mean | SD[f] | P[g] | N[c] | Min[d] | Max[e] | Mean | SD[f] | P[g] |
| PER3 | P415A | UKB | C/C | 23.77 | 39,655 / 127,177 | <0.001 | 33,998 | 23.08 | 31.51 | 27.32 | 0.99 | 0.053 | 33,908 | 23.37 | 30.59 | 27.01 | 0.85 | 0.014 |
| | | | C/G | 29.58 | 463 / 1,102 | | 338 | 21.34 | 31.43 | 27.21 | 1.05 | | 338 | 19.45 | 29.64 | 26.90 | 0.96 | |
| | | | G/G | 42.86 | 3 / 4 | | 1 | 28.01 | 28.01 | 28.01 | - | | 1 | 27.60 | 27.60 | 27.60 | - | |
| | | FINRISK/ Health 2000-2011 | C/C | 22.4 | 613 / 2,121 | 1.000 | - | - | - | - | - | - | - | - | - | - | - | |
| | | | C/G | 22.8 | 34 / 115 | | - | - | - | - | - | | - | - | - | - | - | - |
| | | | G/G | 0.00 | 0 / <5 | | - | - | - | - | - | | - | - | - | - | - | |
| | | MESA | C/C | - | - | - | - | - | - | - | - | - | 1,925 | 13.15 | 34.92 | 27.05 | 2.16 | 0.900 |
| | | | C/G | - | - | | - | - | - | - | | | 10 | 26.12 | 27.98 | 26.96 | 0.67 | |
| | H417R | UKB | A/A | 23.77 | 39,655 / 127,177 | <0.001 | 33,997 | 23.08 | 31.51 | 27.32 | 0.99 | 0.06 | 33,907 | 23.37 | 30.59 | 27.01 | 0.85 | 0.018 |
| | | | A/G | 29.55 | 463 / 1,104 | | 339 | 21.34 | 31.43 | 27.21 | 1.05 | | 339 | 19.45 | 29.64 | 26.90 | 0.96 | |
| | | | G/G | 42.86 | 3 / 4 | | 1 | 28.01 | 28.01 | 28.01 | - | | 1 | 27.60 | 27.60 | 27.60 | - | |
| | | FINRISK/ Health 2000-2011 | A/A | 22.4 | 613 / 2,121 | 1.000 | - | - | - | - | - | - | - | - | - | - | - | |
| | | | A/G | 22.8 | 34 / 115 | | - | - | - | - | - | | - | - | - | - | - | - |
| | | | G/G | 0.00 | 0 / <5 | | - | - | - | - | - | | - | - | - | - | - | |
| | | MESA | A/A | - | | - | - | - | - | - | - | | 1,925 | 13.15 | 34.92 | 27.05 | 2.16 | 0.900 |
| | | | A/G | - | | | - | - | - | - | - | - | 10 | 26.12 | 27.98 | 26.96 | 0.67 | |
| CRY2 | A260T | UKB | G/G | 23.77 | 39,655 / 127,179 | 0.340 | 33,998 | 23.08 | 31.51 | 27.32 | 0.99 | 0.092 | 33,908 | 23.37 | 30.59 | 27.01 | 0.85 | 0.040 |
| | | | G/A | 15.79 | 6 / 32 | | 4 | 27.32 | 28.67 | 28.15 | 0.58 | | 4 | 26.63 | 29.34 | 27.88 | 1.14 | |
| TIMELESS | R1081X | UKB | G/G | 23.77 | 39,652 / 127,175 | 1.000 | 33,997 | 23.08 | 31.51 | 27.32 | 0.99 | - | 33,907 | 23.37 | 30.59 | 27.01 | 0.85 | - |
| | | | G/A | 20.00 | 1 / 4 | | 0 | - | - | - | - | | - | - | - | - | - | |

[a]percentage of individuals in genotype group self-reporting being "definitely a 'morning' person"

[b]Fisher's exact-test P-value (two-sided)

[c]number of individuals

[d]minimum phenotypic value

[e]maximum phenotypic value

[f]SD = standard deviation

[g]carrier status t-test P-value (two-sided). Homozygous carriers for rare alleles combined with heterozygote carriers when performing Fisher's exact-test.

## *CRY1* c.1657+3A>C is associated with chronotype and a delayed sleep phase in a population-based cohort, but with reduced effect size

*CRY1* c.1657+3A>C has previously been associated with delayed sleep phase disorder, characterised by an approximately 1-hour shift towards later sleep and wake times[10]. In the UK Biobank, 10.1% and 11.1% of *CRY1* heterozygous and homozygous variant carriers, respectively, reported being "definitely an evening person", compared to 7.9% of non-variant carriers (Fisher's exact *P* = 0.003) (**Table 5**). No carriers of the *CRY1* variant reported being "definitely an evening person" in the Finnish studies. The observed difference in the UK Biobank in

**Table 5. Proportion of individuals self-reporting as being "definitely an evening person", average L5 timing, and average sleep-midpoint across all nights for each genotype group of the *CRY1* variant previously reported to be causal for delayed sleep phase disorder in the UK Biobank (UKB), Finnish and MESA studies.** Accelerometer based estimates of sleep timing unavailable in the Finnish studies. Self-reported "eveningness" and accelerometer estimates of L5-timing unavailable in MESA.

| Gene | Variant | Study | Genotype | Definitely an "Evening" Person | | | Accelerometer L5 Timing | | | | | | Accelerometer Sleep Midpoint | | | | | |
|---|---|---|---|---|---|---|---|---|---|---|---|---|---|---|---|---|---|---|
| | | | | %[a] | Cases/ Controls | P[b] | N[c] | Min[d] | Max[e] | Mean | SD[f] | P[g] | N[c] | Min[d] | Max[e] | Mean | SD[f] | P[g] |
| *CRY1* | c.1657 +3A>C | UKB | T/T | 7.92 | 13,212 / 153,621 | 0.003 | 33,998 | 23.08 | 31.51 | 27.32 | 0.99 | 0.132 | 33,908 | 23.37 | 30.59 | 27.01 | 0.85 | 0.059 |
| | | | T/G | 10.07 | 149 / 1,331 | | 318 | 24.06 | 30.38 | 27.41 | 1.00 | | 318 | 24.28 | 29.60 | 27.11 | 0.88 | |
| | | | G/G | 11.11 | 1 / 8 | | 4 | 25.08 | 28.44 | 26.50 | 1.48 | | 4 | 25.49 | 27.38 | 26.50 | 0.90 | |
| | | FINRISK/Health 2000-2011 | T/T | 10 | 284 / 2,554 | 1.000 | - | - | - | - | - | - | - | - | - | - | - | - |
| | | | T/G | 0 | 0 / <5 | | - | - | - | - | - | | - | - | - | - | - | |
| | | MESA | T/T | - | - | - | - | - | - | - | - | - | 1,914 | 13.19 | 34.92 | 27.05 | 2.14 | 0.568 |
| | | | T/G | - | - | | - | - | - | - | - | | 21 | 13.15 | 30.10 | 26.78 | 3.32 | |

[a]percentage of individuals in genotype group self-reporting being "definitely a 'evening person"

[b]Fisher's exact-test *P*-value (two-sided)

[c]number of individuals.

[d]minimum phenotypic value

[e]maximum phenotypic value

[f]standard deviation

[g]carrier status t-test *P*-value (two-sided). Homozygous carriers for rare alleles combined with heterozygote carriers when performing Fisher's exact-test.

sleep-midpoint estimated from accelerometery for individuals with a *CRY1* variant was 5.4 minutes later (95% CI:-0.2,11.0, *P* = 0.06), with a similar difference observed for L5 timing. Similar point estimates were observed in our sensitivity analyses that included restricting accelerometer data analyses to either weekend nights or weekday nights. For example, we observed carriers of the *CRY1* variant having a 6.4 minute later sleep-midpoint at weekends (95% CI, 0.4,12.4, *P* = 0.04) (**S9–S11 Tables**).

## Heterozygous protein truncating variants in reported Mendelian sleep genes *PER2* and *PER3* are associated with sleep timing

Most of the reported variants are missense variants and some, for example, *CRY1* c.1657 +3A>C have been shown to have a specific gain of function effect in *in-vitro* experiments[10]. Only the reported *TIMELESS* gene variant is a protein truncating variant (PTV) and it is unclear whether the reported Mendelian sleep variants act through loss of function due to haploinsufficiency. We therefore identified all rare (MAF<0.01%) nonsense, frameshift and essential splice site variants across the ten Mendelian sleep and circadian genes. The number of individuals with rare high confidence loss-of-function variants in these genes ranged from 10 for *ADRB1* to 205 for *TIMELESS*. We subsequently performed burden testing of loss-of-function variants for these 10 genes in up to 184,065 individuals of European ancestry, including and adjusting for relatedness (**S12–S15 Tables**). We identified associations between *PER2* and self-reported measures of chronotype and accelerometer-estimates of sleep timing. We observed associations between loss-of-function variants in the *PER2* gene and UK Biobank participants self-reporting as "definitely a morning person" (Burden *P* = $4 \times 10^{-8}$) (**S13 Table**), and accelerometer-estimates of sleep-midpoint (Burden *P* = $5 \times 10^{-7}$) and L5 timing (Burden *P* = $9 \times 10^{-4}$)(**S15 Table**). Of 64 unrelated European carriers carrying at least one of the 50 loss-of-function variants in the *PER2* gene, 58% (n = 37) self-reported as "definitely a morning person" in contrast to 24% (n = 40,438) among non-carriers (Fisher's exact *P*<0.0001) (**S16**

Table). Within unrelated individuals, compared with non-carriers of loss-of-function variants in *PER2*, carriers had an earlier average sleep-midpoint of ~57 minutes (t-test *P*<0.0001) (S17 Table) and an earlier average L5 timing of ~33 minutes (t-test *P* = 0.027) (S18 Table). In addition, our burden testing identified an association between loss-of-function variants in *PER3* and L5-timing (Burden *P* = 4×10$^{-6}$) (S15 Table). Our gene-based analyses for high confidence PTVs did not result in associations for other previously reported monogenic genes for sleep duration or timing. We observed no associations in gene-based tests of rare missense variants in these genes after accounting for multiple testing (Bonferroni *P* = 0.0001 based on 42 tests across 10 genes).

## Discussion

Recent studies have identified 10 genes where specific variants are reported to cause familial natural short sleep, familial advanced sleep phase, or delayed sleep phase. These studies have tended to be based on a limited number of families ascertained to have a specific sleep trait. This form of ascertainment means the effect of the variants and genes when identified in the population is unknown. Here, we show that most previously reported variants for Mendelian sleep and circadian conditions are not highly penetrant when ascertained incidentally from the general population. Incidental findings are becoming increasingly common with the increase in whole genome sequencing both from direct-to-consumer companies and through health services. It is important, therefore, to get accurate estimates of the risk of developing a condition so that individuals are not misdiagnosed and/or potentially incorrectly treated for a condition.

We and others have shown previously [13] that the penetrance of rare Mendelian disease variants is likely to be lower when estimated from population-based cohorts than in ascertained discovery or clinical cohorts and this may be the case for the genes and variants reported here. It is also possible that some of the reported genes and variants are not causes of the reported sleep conditions in humans. The functional effect of each of the variants assessed here are supported by extensive *in vitro* and animal model studies. The level of human genetic evidence varies across studies, from 6 individuals from a single pedigree for the *ADBR1* variant to 78 individuals from 7 families for the CRY1 c.1657+3A>C variant. While functional evidence in animal models is important to understand the biology of the associated variants, it is important to demonstrate robust human genetic evidence to ensure relevance in humans. It is therefore possible that some of the reported variants with weaker human genetic evidence do not cause monogenic sleep and circadian conditions in humans. However, because of the nature of the ascertainment, our study cannot address pathogenicity and can only conclude that the effect of these variants when identified incidentally from the population appears to be much weaker than previously reported.

There are several other possible explanations for the differences between our studies. Genetic background may play a role. Given the importance of sleep for survival, homeostatic mechanisms are extremely robust. For example, the very strong circadian phenotype of homozygous *PER2* KO allele [27] was completely absent when crossed onto a C57/Bl6 background [28]. Our work underscores the significant challenges of behavioral genetics. It is possible that an individual or family presenting to a clinic with a specific sleep or circadian condition have an increased polygenic susceptibility to sleep duration, in addition to the monogenic variant. For example, we have recently identified 351 variants for being a morning person from genome-wide association studies. Individuals in the highest 5% of polygenic risk had an average sleep timing of 25 mins earlier compared to the lowest polygenic risk individuals. As has been shown for traits such as lipids and BMI [29], this suggests that extreme polygenic risk can

have similar effect sizes to monogenic variants for sleep traits. Ancestry differences may also play a role. For example, the *CRY1* c.1657+3A>C association was discovered in an American family and followed up in Turkish families. A recent paper has found association with the *CRY1* variant and sleep-midpoint timing (~40 mins) in an independent cohort of Turkish ancestry individuals [30]. The difference in effect size between these studies may be due to different genetic backgrounds between these previous populations and the predominantly Northern European ancestry population used in our analyses. There are also potential environmental (e.g. daylight hours) and societal explanations for the different results in this study compared to previous studies.

There are several limitations to our study which provide other possible explanations for the weaker associations observed here compared to previous studies. First, the UK Biobank has a healthy volunteer bias [31] and may select against individuals with sleep disorders. However, this is unlikely for FASP and FNSS where the phenotypes of phase advance or short sleep rarely affect individuals' well-being [20]. Additionally, the allele frequency of the variants in UK Biobank is similar to that in a large resource of exome data (gnomAD) suggesting limited selection against these variants. Second, reported sleep patterns that are often shaped by social factors, and thus may not reflect their underlying sleep preferences. This could explain the lack of association with sleep timing for example. However, we find no association with the variants with traits such as self-reported ease of getting up and limiting the analyses to an individual's activity on the weekend shows a similar lack of association. Additionally, we find no association with any measures of sleep quality or disruption in the UK Biobank, including those defined by medical record codings, although sleep fragmentation has been observed in *CRY1* c.1657+3A>C carriers [10].

Another limitation to this study is that it is not possible to do as detailed sleep and circadian phenotyping in this large-scale study as is possible in smaller scale clinical studies. We have, however, used multiple data sources including self-report and activity monitor data and have used primary care and inpatient data to identify sleep disorders. We have previously validated the activity monitor sleep estimates against polysomnography data [32] and it is a reliable measure of many sleep parameters. The validity of the measures is also confirmed by the statistically robust associations with *PER2* and *PER3* protein truncating variants using both self-report and accelerometry, with effects of sleeping timing of approximately 1 hour. We have also demonstrated this through the robust association of hundreds of common genetic variants with chronotype [14], sleep duration[33] and other sleep measures through genome-wide association studies (GWAS)[19,21]. The number of individuals with variants is relatively low for some genes, but the number of individuals carrying previously identified variants is usually larger than the number of available in the original reports.

Our work demonstrates that haploinsufficiency of *PER2* affects circadian timing in humans. We find a substantial effect on chronotype and sleep timing for individuals with heterozygous *PER2* protein truncating variants. This is unexpected because only homozygous *Per2* knockout mice exhibit a circadian shortened circadian period, with no phenotype in heterozygotes [34]. The human genetic evidence for a role of the *PER2* S662G missense variant is robust with co-segregation in a large pedigree with FASPS [5]. It was initially thought that the effect of this S662G variant was caused by decreased phosphorylation of PER by CK1ε that could stabilize it leading to PER accumulating prematurely and shortened circadian period. Others have suggested that the S662G mutation results in decreased PER2 levels and/or an increased turnover of nuclear PER2 [35]. Our work shows that, in humans, haploinsufficiency of *PER2* causes a substantial effect on chronotype and sleeping timing.

Our results indicate that most previously reported variants for Mendelian sleep and circadian conditions are not highly penetrant causes of extreme sleep duration or timing when ascertained incidentally from the population.

## Supporting information

**S1 Table. Summary of twelve variants previously reported to be causal for Mendelian sleep and circadian conditions, including the variant frequencies catalogued in gnomAD.**
(DOCX)

**S2 Table. Maximum genotype counts for 12 previously reported monogenic causes of sleep and circadian conditions in unrelated individuals of European ancestry from the UK Biobank, FINRISK / Health 2000-2011, and MESA studies.** Genotype counts are based on availability of sleep characteristics relevant to each gene.
(DOCX)

**S3 Table. Summary statistics of self-reported sleep duration in the UK Biobank (Field 1160) for carriers of variants previously described as causal for familial natural short sleep.**
(DOCX)

**S4 Table. Summary statistics of dichotomised self-reported sleep data in the UK Biobank for carriers of variants previously described as causal for familial natural short sleep.** Data unavailable for self-reported sleep of $\leq 5$ hours, $\leq 4$ hours and 4–6 hours in the Finnish study.
(DOCX)

**S5 Table. Summary statistics of accelerometer-derived estimates of sleep duration in UK Biobank for carriers of variants previously described as casual for familial natural short sleep.** No carriers of the *GRM1* A889T variant remained among individuals from UK Biobank who had worn an accelerometer.
(DOCX)

**S6 Table. Summary statistics of "Morningness" across genotype groups for variants previously reported as causal for familial advanced sleep phase.** Data on being "more or definitely a morning person" unavailable in the Finnish studies.
(DOCX)

**S7 Table. Summary statistics of L5-midpoint timing estimated from accelerometer data in UK Biobank across genotype groups for variants previously reported as causal for familial advanced sleep phase.**
(DOCX)

**S8 Table. Summary statistics of sleep-midpoint estimated from accelerometer data in UK Biobank and MESA across genotype groups for variants previously reported as causal for familial advanced sleep phase.**
(DOCX)

**S9 Table. Summary statistics of "eveningness" across genotype groups for variants previously reported as causal for delayed sleep phase.** Data on being "more or definitely an evening person" unavailable in the Finnish studies.
(DOCX)

**S10 Table. Summary statistics of L5-midpoint timing estimated from accelerometer data in UK Biobank a across genotype groups for variants previously reported as causal for delayed sleep phase.**
(DOCX)

**S11 Table. Summary statistics of sleep-midpoint estimated from accelerometer data in UK Biobank and MESA across genotype groups for variants previously reported as causal for**

**delayed sleep phase.**
(DOCX)

**S12 Table. P-values from burden testing of rare (MAF < 0.01%) loss-of-function and missense variants in genes outlined in this paper on self-reported sleep duration in UK Biobank.**
(DOCX)

**S13 Table. P-values from burden testing of rare (MAF < 0.01%) loss-of-function and missense variants in genes outlined in this paper on chronotype in UK Biobank.**
(DOCX)

**S14 Table. P-values from burden testing of rare (MAF < 0.01%) loss-of-function and missense variants in genes previously reported to harbour variants causal for disruptive sleep duration or timing on accelerometer estimates of sleep duration in UK Biobank.** There were no remaining loss-of-function carriers for *GRM1*, *ADRB1* and *CRY2* within the subset of individuals from UK Biobank who wore an accelerometer.
(DOCX)

**S15 Table. P-values from burden testing of rare (MAF < 0.01%) loss-of-function and missense variants in genes previously reported to harbour variants causal for disruptive sleep duration or timing on accelerometer estimates of sleep timing in UK Biobank.** There were no remaining loss-of-function carriers for *GRM1*, *ADRB1* and *CRY2* within the subset of individuals from UK Biobank who wore an accelerometer.
(DOCX)

**S16 Table. Summary of chronotype by PER2 loss-of-function carrier status in the UK Biobank.**
(DOCX)

**S17 Table. Summary of sleep-midpoint by PER2 loss-of-function carrier status in the UK Biobank.**
(DOCX)

**S18 Table. Summary of L5-midpoint timing by PER2 loss-of-function carrier status in the UK Biobank.**
(DOCX)

**S1 Methods. Supplementary Methods.** Description of Additional Studies.
(DOCX)

## Acknowledgments

We thank Louis Ptáček, Alina Patke and Michael Young for helpful discussion and comments on the manuscript. We gratefully acknowledge the studies and participants who provided biological samples and data for MESA and TOPMed. The Finnish data used for the research was obtained from THL Biobank (study number BB2019_43). We would like to thank all study participants for their generous participation at THL Biobank and cohorts FINRISK 1992, 1997, 2002, 2007, and 2012, as well as Health 2000 and 2011. We want to acknowledge the participants and investigators of the FinnGen study.

## Author Contributions

**Conceptualization:** Michael N. Weedon, Andrew R. Wood.

**Data curation:** Samuel E. Jones, Jess Tyrrell, Robin N. Beaumont, Timo Partonen, Ilona Merikanto, Andrew R. Wood.

**Formal analysis:** Michael N. Weedon, Samuel E. Jones, Jiwon Lee, Amy Dawes, Andrew R. Wood.

**Resources:** Hanna M. Ollila.

**Supervision:** Michael N. Weedon, Tamar Sofer, Andrew R. Wood.

**Writing – original draft:** Michael N. Weedon, Samuel E. Jones, Richa Saxena, Andrew R. Wood.

**Writing – review & editing:** Michael N. Weedon, Samuel E. Jones, Jacqueline M. Lane, Hanna M. Ollila, Amy Dawes, Robin N. Beaumont, Stephen S. Rich, Jerome I. Rotter, Timothy M. Frayling, Martin K. Rutter, Susan Redline, Tamar Sofer, Richa Saxena, Andrew R. Wood.

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
