## [Decision Letter · Decision Letter 0]

3 May 2022

Dear Dr Wood,

Thank you very much for submitting your Research Article entitled 'The impact of Mendelian sleep and circadian genetic variants in a population setting' to PLOS Genetics.

The manuscript was fully evaluated at the editorial level and by two independent peer reviewers. As you will see, both reviewers are positive, but there are some concerns that we ask you address in a revised manuscript

We therefore ask you to modify the manuscript according to the review recommendations. Your revisions should address the specific points made by each reviewer.

In addition we ask that you provide a detailed list of your responses to the review comments and a description of the changes you have made in the manuscript.

[LINK]

Yours sincerely,

Gregory S. Barsh

Editor-in-Chief

PLOS Genetics

Gregory Copenhaver

Editor-in-Chief

PLOS Genetics

Reviewer's Responses to Questions

**Comments to the Authors:**

Reviewer #1: I very much enjoyed reading the authors’ manuscript. It was nice study into the effects of previously found genetic variants linked to sleep and circadian genes. Whilst slightly limited in scope, it is nevertheless an important piece of research.

There were a few things in particular that I very much enjoyed in the paper. Mostly I found the paper easy to read and understand with only a few confusing sections (see below). Verification of the results from the main cohort (UK Biobank) in the two Finnish studies was nice to see and really helps to back up the mostly null results presented in the manuscript. Lastly the detailed sensitivity analysis was very useful in understanding the effects and showing that results presented were not caused by cherry picking of thresholds etc.

That being said I did have a few criticisms of the manuscript, these are mostly minor comments/changes.

• Sleep timing – I feel that “diurnal preference” and “chronotype” are used interchangeably here, with only the former being defined. I would either stick to one, or define both.

• Line 256: redundant word “tests”

• Confusing sub headings. In the Methods section subheadings are in mix of italics and underlined but no consistently. Later in the Results subheadings are just in italics. I felt that the lack of consistency was confusing.

• I don’t think that the logistic regression analyses where presented in full anywhere. As far as I could tell they were given in select cases, the full results from this for all analyses run should be presented somewhere (probably in supplemental).

• Tables 1 and 2 didn’t seem very interesting to me. For the first section in the results as far as I understood the most important analysis was in Sup Table 1. I’d recommend moving T1&2 to supplemental and promoting ST1 to the manuscript.

• ST2: Is the 2-sided t-test correct here? If you’re looking for evidence that individuals have a short sleep, should it not be a 1-sided test? I think similar arguments can be made for later tables as well.

• Line 347: “We previously tested the …” Is this missing a reference here? Likewise on Line 363 “has previously been associated with…” also seems to missing a reference. Either needs references adding in or these sentences need removing.

• Table 5 seems to show an association between sleep midpoint and the CRY2 variant in the accelerometer data, but this isn’t mentioned in the text?

• ST5: Is the “definitely a morning person” section of this table the same as the “definitely a morning person” section of Table 5? If so then why are the Finnish values different in these two tables? Also the p<0.0001 should match the p<0.001 in Table 5 for consistency.

• I would switch the order of ST6 and ST7 (and likewise the order of ST9 and ST10) to match the ordering in Table 5 (and Table 6).

• I think that it’s worth mentioning the observed differences between weekday and weekend nights in ST9

• Supplemental Figures 1-3 add nothing to the manuscript as they are not referred to (just tacked on to line 374). They should be removed.

Reviewer #2: This is a straightforward publication that helps clarify the literature. This is an important and well-done publication. Non-replications, including from the same author’s lab where the work has been initially generated, are important as it is harder and harder to make sense of the literature.

The authors show lack of replication of large effects of 8/10 previously published “mendelian circadian and sleep variants” reported in single or rare families. The authors also extend the study of these genes by conducting burden tests, often finding minimal or no effects, except for PER2 and PER3. It is then argued that out of all 10 mutations previously reported maybe only one, S662G, not found in this sample, maybe pathogenic. The other previously published mutation not found in this survey was NPSR1 Y206H.

I have a few comments. It would be useful to list in the abstract the two out of 10 mutations that were not found (NPSR1 Y206H and PER2 S662G).

The publication focusses mostly on circadian and short sleep variants, so the title is misleading. For example, a new mutation g.42184347T>C; p.Lys68Arg; rs537376938 in the cleavage site of HCRT has been recently reported to be associated with idiopathic hypersomnia (minor allele frequency of 1.67% in cases versus 0.32% in controls, P = 2.7 × 10-8, odds ratio = 5.36) characterized by sleepiness and excessive sleep in Asians (Miyagawa et al. NPJ Genom Med. 2022 Apr 12;7(1):29). This was somewhat surprising as a mouse model with a truncated form of pre-proorexin generating only orexin A, even when homozygous, has no phenotype. Miyagawa et al. reports in the publication that the variant is present in non-Finnish Europeans at 0.013%, so most likely it would be present in the UK biobank analyzed here. Perhaps analyzing this variant should be confirmed and included, as the UK biobank has questions on excessive sleepiness and sleep time. In this case, the title would be more accurate.

The main issues that could have led to a difference in results include: 1) ascertainment of phenotypes, 2) amount of cases/nature of pedigree (the more polygenic a trait is the faster genetic risk decreases from one generation to the next) and 3) genetic background/ethnicity (even so MESA was included, it is a small sample). These limitations are not all well discussed. Even in the small number of cases identified with some of these mutations, a PCA analysis could reveal unsuspected population stratification or founder effects. Has this been looked at?

The fact many of these variants had thorough functional characterization (often in mouse) that was used as evidence of involvement is not discussed at all. The authors need to discuss briefly the limits of functional characterization. Indeed, it is argued that PER2 S662G has sufficient evidence for being pathogenic. Indeed, Weeldon et al show here that haploinsufficiency of PER2 is associated with a morning phenotype. PER2 knock out are also phase advanced (Nature; 1999 Jul 8;400(6740):169-73). Finally, for S662G, there is multiple in vitro evidence linking the S662 phosphorylation site with casein kinases (although the exact mechanism is somewhat debated see Cell. 2007 Jan 12;128(1):22-3. doi: 10.1016/j.cell.2006.12.024), other key circadian genes, plus the effects of S662G and S662A were assessed in mice. This is reasonable.

However, there is also similar rodent evidence be written for other mutations that was used to “confirm” the phenotype that was not confirmed in this study such 1) the T44A in the human CKIδ gene had a lot of experiment support; 2) the A260T of CRY2 which is in its FAD binding domain (postulated to increase the affinity of FAD for the E3 ubiquitin ligase FBXL3, thus promoting its degradation). 3) the Patke et al. CRY1 DSPD mutation, located in the 5′ splice site of exon 11 and leading to exon 11 and an in-frame deletion of 24 residues in the C-terminal region of CRY1. This was postulated to lead to an enhanced affinity of this repressor for the circadian activator proteins CLOCK and BMAL1, which could lengthen the period of circadian molecular rhythms. These also all had reasonable explanations for the phenotype observed. This raises questions on the value of functional characterizations done in mouse or other models that should be briefly mentioned.

Similarly, NPSR1 Y206H which is not found is not discussed. Is there something special about the NPSR1 variant or this gene?

**Have all data underlying the figures and results presented in the manuscript been provided?**

Reviewer #1: Yes

Reviewer #2: Yes

PLOS authors have the option to publish the peer review history of their article (what does this mean?). If published, this will include your full peer review and any attached files.

Reviewer #1: **Yes: **Robert Maidstone

Reviewer #2: No

---

## [Decision Letter · Decision Letter 1]

26 Jul 2022

Dear Dr Wood,

We are pleased to inform you that your manuscript entitled "The impact of Mendelian sleep and circadian genetic variants in a population setting" has been editorially accepted for publication in PLOS Genetics. Congratulations!

The revised manuscript was seen by the previous reviewers, both of whom are enthusiastic as you will see below.

Yours sincerely,

Gregory S. Barsh

Editor-in-Chief

PLOS Genetics

Gregory Copenhaver

Editor-in-Chief

PLOS Genetics

Comments from the reviewers (if applicable):

Reviewer's Responses to Questions

**Comments to the Authors:**

Reviewer #1: I thank the authors for their response to my previous comments. I feel that they have sufficiently answered all of my points and I'm happy with the resulting manuscript. I look forward to seeing it published.

Reviewer #2: The revised version addresses all my comments.

**Have all data underlying the figures and results presented in the manuscript been provided?**

Reviewer #1: Yes

Reviewer #2: Yes

PLOS authors have the option to publish the peer review history of their article (what does this mean?). If published, this will include your full peer review and any attached files.

Reviewer #1: **Yes: **Robert Maidstone

Reviewer #2: No

**Data Deposition**

http://datadryad.org/submit?journalID=pgenetics&manu=PGENETICS-D-22-00319R1

**Press Queries**

---

## [Editor Report · Acceptance letter]

23 Aug 2022

PGENETICS-D-22-00319R1 

The impact of Mendelian sleep and circadian genetic variants in a population setting 

Dear Dr Wood, 

We are pleased to inform you that your manuscript entitled "The impact of Mendelian sleep and circadian genetic variants in a population setting" has been formally accepted for publication in PLOS Genetics! Your manuscript is now with our production department and you will be notified of the publication date in due course.

With kind regards,

Livia Horvath

PLOS Genetics

On behalf of:
